

# Application of built-in adjuvants for epitope-based vaccines

Yao Lei[1,2], Furong Zhao[1,2], Junjun Shao[1,2], Yangfan Li[1,2], Shifang Li[1,2], Huiyun Chang[1,2] and Yongguang Zhang[1,2]

[1] State Key Laboratory of Veterinary Etiological Biology, OIE/National Foot-and-Mouth Disease Reference Laboratory, Lanzhou Veterinary Research Institute, Chinese Academy of Agricultural Sciences, Lanzhou, China
[2] Jiangsu Co-Innovation Center for Prevention and Control of Important Animal Infectious Diseases and Zoonoses, Yangzhou, China

## ABSTRACT

Several studies have shown that epitope vaccines exhibit substantial advantages over conventional vaccines. However, epitope vaccines are associated with limited immunity, which can be overcome by conjugating antigenic epitopes with built-in adjuvants (e.g., some carrier proteins or new biomaterials) with special properties, including immunologic specificity, good biosecurity and biocompatibility, and the ability to vastly improve the immune response of epitope vaccines. When designing epitope vaccines, the following types of built-in adjuvants are typically considered: (1) pattern recognition receptor ligands (i.e., toll-like receptors); (2) virus-like particle carrier platforms; (3) bacterial toxin proteins; and (4) novel potential delivery systems (e.g., self-assembled peptide nanoparticles, lipid core peptides, and polymeric or inorganic nanoparticles). This review primarily discusses the current and prospective applications of these built-in adjuvants (i.e., biological carriers) to provide some references for the future design of epitope-based vaccines.

## INTRODUCTION

Vaccination is a major preventive measure designed to establish specific immune defenses (i.e., antibody or cellular immunity) to protect individuals from infectious diseases. In 1796, the British rural doctor, Edward Jenner, conducted a scientific study on the prevention of smallpox in humans and demonstrated that vaccination with vaccinia virus could prevent smallpox, from which the terms vaccinology and immunology originated (*Negahdaripour et al., 2017b*). Traditional vaccines typically include inactivated or attenuated vaccines derived by reducing the virulence of the pathogen by physical or chemical methods (*Skwarczynski & Toth, 2011a*; *Karch & Burkhard, 2016*). Due to the continuous progress of science and technology (i.e., immunology and molecular biology), subunit vaccines based on short, specific pathogen fragments have undergone increased development to compensate for the shortcomings of traditional vaccines, including low biosafety (reversion to virulence), inefficient cultivation of pathogens, and the occurrence of allergies and autoimmunity (*Skwarczynski & Toth, 2014*).

Corresponding authors
Huiyun Chang,
changhuiyun@caas.cn
Yongguang Zhang,
zhangyongguang@caas.cn

Moreover, epitope-based vaccines play an important role in current vaccine research and exhibit several advantages over conventional vaccines, including high specificity, good safety, ease of production and storage, and stability. As a result of these advantages, epitope-based vaccines have become an area of growing interest in the field of vaccine research (*Skwarczynski & Toth, 2014*; *Hajighahramani et al., 2017*; *Nezafat et al., 2016*, *2017*).

Since antigenic peptides are easily degraded by proteases in the body, it is difficult for the receptors expressed on the immune cells to identify antigen epitopes, and they do not generate a strong immune response to pathogens. An epitope-based vaccine with a reasonable design is composed of epitope peptide/s, a delivery system, and an adjuvant (*Rueckert & Guzman, 2012*). For multi-epitope vaccines, since the traditional carriers and adjuvants are associated with poor efficacy, vaccine designs with built-in adjuvants have been proposed. Therefore, a built-in adjuvant exhibiting both the functions of a transmission system and a traditional adjuvant, is constructed within the vaccine to improve the immunogenicity of epitope peptides by stimulating the innate immune response required for an adaptive immune response. To achieve this goal, the epitopes are regularly fused with adjuvant proteins (e.g., toll-like receptor (TLR) ligands and proteins that can spontaneously assemble into virus-like particles (VLPs)) or displayed on the surface of some particular biomaterials (e.g., liposomes, gold nanoparticles, and poly(lactic-*co*-glycolic acid) (PLGA)) and the immunogenicity of the epitopes are significantly increased by this immune complex (*Chen et al., 2017*; *Rueda et al., 2017*; *Kitaoka et al., 2017*; *Karuturi et al., 2017*). This review primarily introduces the methods for applying built-in adjuvants in the design of epitope-based vaccines, including a few new delivery systems (e.g., dendrimers, self-assembled peptide nanoparticles (SAPNs), and hyperbranched polyglycerol (hbPG)) (*Busseron et al., 2013*; *Glaffig et al., 2015*; *Indelicato, Burkhard & Twarock, 2017*).

## SURVEY METHODOLOGY

In this paper, we reviewed articles related to the built-in adjuvants of epitope-based vaccines. All references in this review paper were retrieved using search engines, such as PubMed, Google Scholar. Keywords, including epitope vaccine, built-in adjuvants, biological carriers, and nanoparticles (NPs) were used to search for relevant references.

## MAJOR IMMUNOLOGICAL CONCEPTS OF EPITOPE-BASED SUBUNIT VACCINES

Epitope-based subunit vaccines are typically composed of multiple epitopes derived from one or more pathogenic microorganisms (*Azmi et al., 2014*; *Nezafat et al., 2017*). These epitopes are generally composed of B cell epitopes, cytotoxic T cell (CTL) epitopes, and helper T cell (Th) epitopes. B cells identify thymus-dependent antigen through B-cell receptors (BCRs) expressed on their surface. The activation of B cells and the transfer of signals following activation requires two signals: (1) the BCR-specific receptor directly identifies B cell epitopes of the pathogen; and (2) the interaction between multiple
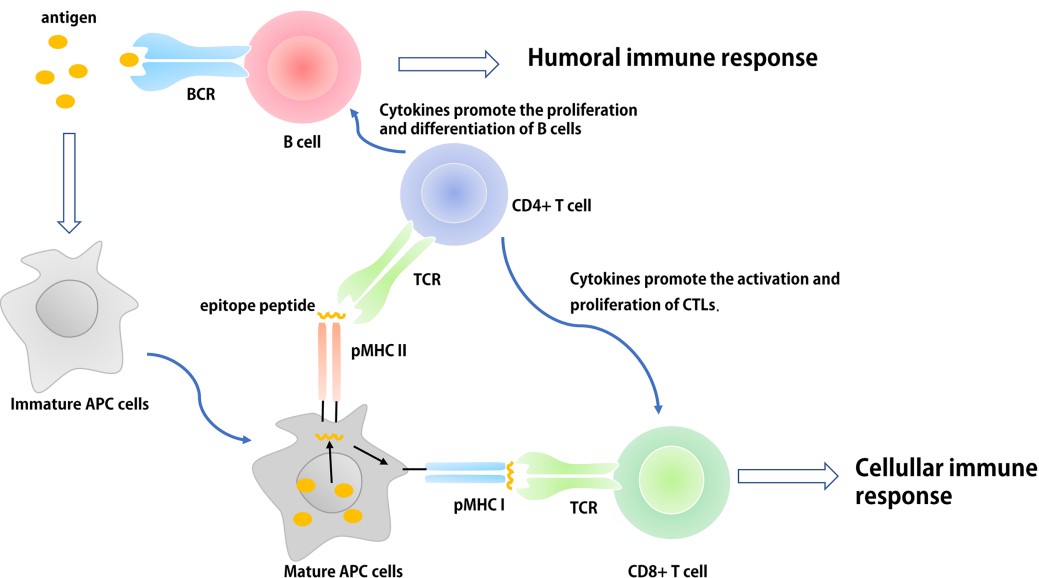

**Figure 1 The basic process of immune response in vivo.** The antigen is ingested and processed by immature antigen-presenting cells (e.g., DCs), APCs becomes mature under the action of immune-stimulating molecules. Mature APCs can express antigen information on its surface in the form of antigen peptide-MHC molecular complex and present it to T cells. After recognizing this complex, T cells are activated, proliferated, and differentiated into different subtypes of effector T cells (CD4[+] and CD8[+]) to participate in the regulation of antigen-specific humoral and cellular immune responses.

co-stimulators on the surface of Th cells and B cells. In addition, activated B cells expressing a variety of cytokine receptors can proliferate in response to cytokines secreted by activated Th cells. The interaction between the specific T cell receptor and the antigen peptide-major histocompatibility complex is known as antigen recognition, which is the first signal required for T cell activation. The interaction between many of the co-stimulatory molecules expressed on the surface of T cells and antigen presentation cells (APCs) (e.g., dendritic cells (DCs)) facilitates the complete activation of T cells. Activated Th1 cells secrete a variety of cytokines (e.g., IL-2, TNF-β, and IFN-γ) some of which (e.g., IL-2) induce non-professional or professional APCs to express co-stimulatory molecules that provide the second signal for the activation of CTLs (*Moyle & Toth, 2013*; *Skwarczynski & Toth, 2014*). These cytokines can also promote the activation and proliferation of Th1, Th2, CTL, and natural killer cells, and expand the cellular immune response. Th2 cells further promote the proliferation and differentiation of B cells and assist the humoral immune response by producing cytokines (e.g., IL-4, IL-5, IL-10, and IL-13) and establishing CD40-CD40L connections with B cells (Fig. 1). However, the individual specific epitopes of pathogenic microorganisms are often unable to induce adequate CTL and antibody responses due to a lack of appropriately activated Th cells and pathogen-derived molecules. Therefore, when designing epitope-based vaccines, researchers typically concatenate antigen-specific B cell or CTL epitopes with Th-cell epitopes with appropriate flexible linkers (e.g., GPGPG and EAAK) (*Wang et al., 2011*, *2018a*; *Nezafat et al., 2015*).

**Table 1  Different subtypes of TLRs and their identified PAMPs.**

| TLRs | PAMPs | Biological activity | Reference |
|---|---|---|---|
| TLR2/TLR6 TLR2/TLR1 | The lipoproteins of bacteria or mycoplasma. Lipopeptide (MALP-2), Peptidogl, ycan (PGN) | Activate intracellular signal NF-KB, induce adhesion molecules and inflammatory cytokines. | Zhu et al. (2010), Basto & Leitao (2014), Kaisho & Akira (2002) |
| TLR4 | Lipopolysaccharides(LPS), Heat shock protein (HSP), β-defensin, Heparin-binding hemagglutinin(HBHA) | Induce the expression of adhesion molecules and inflammatory cytokines. | Reed et al. (2016), Kaisho & Akira (2002) |
| TLR5 | Gram-negative bacteria flagellin | The potent proinflammatory activity by inducing NF-KB activation, and expression of IL-8 and inducible NO synthase in intestinal epithelial cells. | Kaisho & Akira (2002), Moyle (2017) |
| TLR3 | Double-stranded RNA (ds RNA), Poly(I:C) | Induce IL-12 production and DC maturation and elevate CD40 expression on APCs. | Kaisho & Akira (2002), Cheng et al. (2018) |
| TLR7/TLR8 | Single-stranded RNA (ssRNA) | Induce the expression of adhesion molecules and inflammatory cytokines | Vasilakos & Tomai (2013) |
| TLR9 | CpG DNA, Hemozoin, Herpes simplex virus DNA | Production of Th1 cytokines and promotion of cytotoxic activity of NK cells. | Zhu et al. (2010), Kaisho & Akira (2002) |

**Note:**
The biological activities of TLRs agonists that can activate the immune system.

Innate immune cells (e.g., monocyte-macrophages and DCs) recognize different pathogens through pattern recognition receptors (PRRs). Their antigen-presenting and cytokine regulation effects initiate adaptive immune responses, influence the intensity and type of an adaptive immune response, and participate in the generation of immunological memory. In the design of epitope-based vaccines, researchers generally regard the application of built-in adjuvants as an important platform used to provide suitable Th-cell epitopes for specific pathogens or pathogen related molecular patterns (PAMPs) to activate the innate immune response. Thus, some molecular adjuvants or carriers with no infectious and toxic components can be used as built-in adjuvants to facilitate the presentation of pathogen epitopes to the immune system (Moyle & Toth, 2013; Foged, 2011; Shirbaghaee & Bolhassani, 2016).

# TOLL-LIKE RECEPTOR AGONISTS AS BUILT-IN ADJUVANTS

The development of immunotherapeutic vaccines based on T cell immune responses is essential for the prevention and control of cancer and viral diseases. To achieve this goal, researchers must identify a built-in adjuvant that can stimulate strong Th1 immune responses (Felzmann, Gadner & Holter, 2002). One of the strategies for designing epitope-based vaccines is to use TLR ligands as adjuvants that can polarize CD4$^+$ Th cells and induce CTL responses (Van Der Burg et al., 2006). TLRs are a type of PRR that can both induce an innate immune response and activate the adaptive immune system following PAMP activation (Allison, Benoist & Chervonsky, 2011). To date, researchers have discovered 10 TLRs, termed TLR1–TLR10 (Table 1). TLR ligands are expressed by different microorganisms; for example, bacteria harbor a variety of TLR

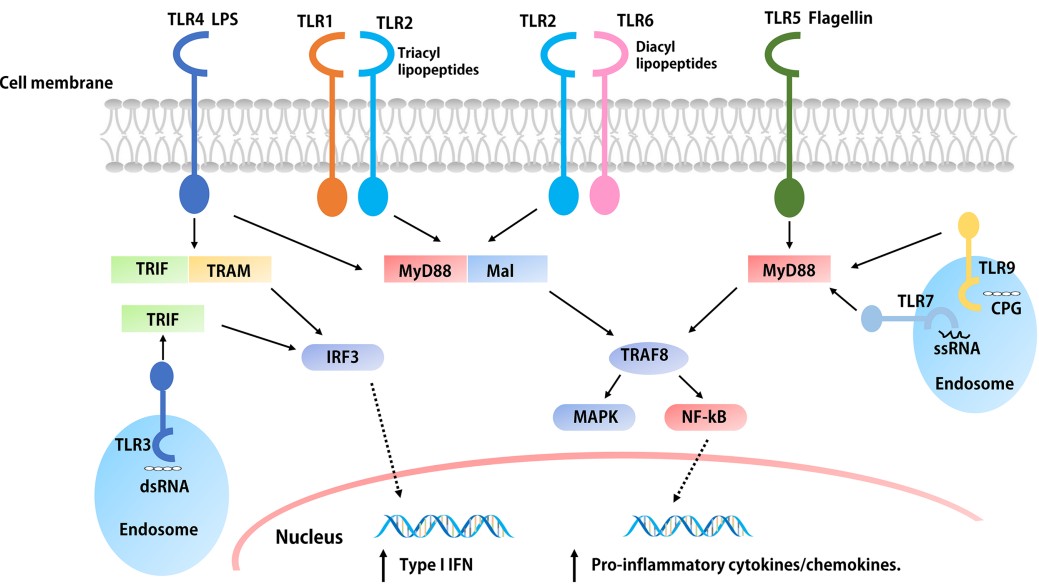

**Figure 2 The signaling pathways of TLRs.** The extracellular parts of TLRs are activated after binding with ligands, and the conformation changes lead to convergence of downstream molecules, which triggers the signaling pathway and induced the up-regulation and activation of cytokines, chemokines, and other co-stimulatory factors. With the exception of TLR3, all TLRs initiate MyD88 through the expressed MyD88 or simultaneous bridging MAL, and then activate the NF-kB and MAPK through tandem reactions, which induces the production of pro-inflammatory cytokines such as IL-1, IL-6, TNF-α, etc. The overexpression of both TRIF and TRAM or TRIF alone initiated the TRIF dependent pathway, the TRIF dependent pathway activates IFN regulatory factors and mediates the production of type I IFNs. In addition, the activation of TLR4 is related to both pathways.

ligands, including TLR2 (macrophage-activating lipoprotein 2 (MALP-2)); TLR4 (e.g., LPS, HSP, and HBHA), TLR5 (e.g., flagellin), and TLR9 (CpG DNA) ligands (*Zhu et al., 2010*). The role of TLR10 is to inhibit rather than activate the immune system, and its ligands are poorly understood (*Jiang et al., 2016*). The extracellular portion of the TLR is activated after binding to an appropriate ligand, and the conformational changes cause convergence of downstream molecules, which triggers associated signaling pathways and induces the up-regulation and activation of cytokines, chemokines, and other costimulatory factors (*Dowling & Dellacasagrande, 2016*) (Fig. 2). Thus, TLR ligands are promising candidates for the development of novel built-in vaccine adjuvants, which can enhance the immunogenicity of epitope-based vaccines. According to the different biological characteristics of various TLRs, appropriate PAMPs are often selected as the molecular binding proteins of epitope peptides to be used as immune adjuvants (*Basto & Leitao, 2014*). In this review, several TLR ligands that are often used as built-in adjuvants for epitope vaccines are introduced.

## Lipopeptides

As a lipopeptide derived from mycoplasma and a potential agonist of TLR2/6 heterodimers in DCs (*Takeda et al., 2018*), MALP-2 is the most widely studied TLR2 adjuvant and has attracted great interest as a novel and efficient built-in adjuvant for

vaccines against infectious diseases (*McDonald et al., 2014*; *Muhlradt et al., 1997*). MALP-2 has been used in phase I and II trials involving pancreatic cancer patients (*Schmidt et al., 2007*). *McDonald et al. (2014)* constructed a variety of self-adjuvating MUC1-MALP-2 conjugate vaccine candidates and demonstrated that the vaccine candidates could induce a high level of humoral immunity without the participation of an external adjuvant and Th epitopes in animal models. Fibroblast stimulating lipopeptide-1 (FSL-1; Pam2CGDPKHPKSF) and synthetic lipopeptide, Pam2CSK4, derived from the LP44 lipoprotein of *Mycoplasma salivarium* can activate macrophages as a TLR2/TLR6 ligand (*Kurkjian et al., 2017*; *Liu et al., 2016*). *Liu et al. (2016)* used FSL-1 as a built-in adjuvant and synthesized a new type of self-adjuvating (glyco) lipopeptide cancer vaccine. Recently, the TLR2/6 ligand, Pam2CSK4, was used as a Th2 polarizing adjuvant in the study of *Leishmania* major and *Brugia malayi* murine vaccines (*Halliday et al., 2016*). In addition, Pam2CSK4 is also used as an adjuvant for major outer membrane protein antigen of *Chlamydia trachomatis* to stimulate a robust immune response and induce effective protection against *C. muridarum* (*Cheng et al., 2014a*). The Braun *Escherichia coli* lipoprotein is a prototype composed of the three acylated lipoproteins from the outer membrane of gram-negative bacteria, and some of its synthetic lipopeptides act as TLR2 stimulators (e.g., Pam3CSK4) (*Basto & Leitao, 2014*; *Arai, Inuki & Fujimoto, 2018*). *Cai et al. (2017)* designed and synthesized an HIV-1 glycopeptide immunogen containing a V3 glycan-dependent neutralizing epitope, a universal T helper epitope (P30), and a lipopeptide (Pam3CSK4). After administering the glycopeptide immunogen to rabbits without any additional adjuvants, a glycan-dependent antibody can be produced in a short period of time, and the induced serum antibodies can recognize a wide-range of HIV-1 gp120s across different clades (*Cai et al., 2017*). Pam2Cys and Pam3Cys are also molecular adjuvants used in vaccines (*Zaman & Toth, 2013*; *McDonald, Byrne & Payne, 2015*; *Nalla et al., 2015*) (Table 2; Fig. 3). Synthetic lipopeptides and their analogues play an important role in the study of built-in adjuvants for epitope-based vaccines. However, compared with mature protein carriers for conjugated vaccines, TLR ligands (e.g., lipopeptides and monophosphoryl lipid A derivatives) are still in their early stages as potential vaccine carriers (*Li & Guo, 2018*). However, since their interaction with TLRs has been thoroughly studied and understood, the prospective development of epitope-based vaccines using TLR ligands as built-in adjuvants is foreseeable. At present, one of the major challenges is obtaining a sufficient number of vaccines for clinical research since the structure of these lipopeptides, as well as their synthesis and binding to target antigens is highly complex. Semi-total and aminoalkyl glucosaminide 4-phosphates (AGPs) synthesis are important methods that can be used to solve this problem (*Li & Guo, 2018*; *Persing et al., 2002*). AGPs have a less-complex structure that allows for synthesis to be easier, more efficient, and elicit immunostimulatory activity in preclinical studies. For example, a Hepatitis B vaccine based on AGP has been approved for use in Argentina (*Dupont et al., 2006*). A second challenge is how to make full use of all aspects of the multicomponent vaccine structure through reasonable considerations in vaccine design, which can be optimized through structure-activity relationship analysis and molecular modeling (*Jin et al., 2007*).

Table 2 Various lipopeptides that can be used as build-in adjuvants.

| Name | Natural analogues | Biological activity or structure | Reference |
|---|---|---|---|
| MALP2 | The M161Ag lipoprotein of Mycoplasma fermentans | The agonistic ligand of the TLR2/6 heterodimer. Induces production of inflammatory cytokines from macrophages, monocytes and DCs. MALP2s, a short form of MALP2, lacks the last eight amino acids of the full length MALP2 (Pam2-CGNNDESNISFKEK). As an adjuvant capable of inducing DC maturation, MALP2s can be used in antitumor immunotherapy. | Takeda et al. (2018), McDonald et al. (2014) |
| FSL-1 | The LP44 lipoprotein from Mycoplasma salivarium | FSL-1 (Pam2CGDPKHPKSF) contains the structure of diacylglycerol similar to Pam2CSK4, which play a key role in immune cell maturation and Th2 immunization and induces the expression of inflammatory cytokines, such as monocyte chemotactic protein (MCP)-1, IL-6, IL-8 and tumor necrosis factor (TNF)-α by monocytes/macrophages. | Liu et al. (2016), Kurkjian et al. (2017) |
| Pam3CSK4 | The Braun lipoprotein in Escherichia coli | Pam3CSK4 is the first mimicking lipopeptide that contains three highly lipophilic tails and six amino acids and can activates the TLR2/1 signaling pathway. | Basto & Leitao (2014), Arai, Inuki & Fujimoto (2018) |
| Pam2CSK4 | The LP44 lipoprotein from Mycoplasma salivarium | The palmityl tail on the N-terminal of cysteine of Pam3CSK4 has been shown to be an dispensable part of TLR2 activation. Removing this lipophilic tail forms a highly effective Pam2CSK4. | Halliday et al. (2016), Arai, Inuki & Fujimoto (2018) |
| Pam3Cys | The Braun lipoprotein in Escherichia coli | Modulation of APC Migration and Antigen Internalization. More efficient than CpG and resiquimod (TLR9 andTLR7/8 ligands). The enantiopure Pam3Cys derivatives that contained R-configured glycerol can induce cytokines and antibody production in mice when administered with antigens. The antigen-specific CTL cells induced by S-epimers were significantly higher in mice. Vaccines containing Pam3Cys can reduce the burden of breast tumors in mice and induce the production of CTLs. | Zaman & Toth (2013), McDonald, Byrne & Payne (2015) |
| Pam2Cys | Cytoplasmic membrane of Mycoplasma fermentans | Compared with Pam3Cys, Pam2Cys have higher solubility characteristics and is a more potential stimulus factor splenocytes and macrophages. The activity of the natural R isomer of Pam2Cys is 100 times that of S isomer. Dependent on the palm acylated cysteine lipid head group activates downstream signals and activate TLR2 on DC's and trigger maturation of DCs. | Zaman & Toth (2013), Nalla et al. (2015) |

Note:
The biological activities or structures of various lipopeptides and their natural analogues.

## Heat shock proteins

Heat shock proteins (HSPs) are a type of cellular companion protein produced by biological cells that are stimulated by environmental stressors and can be divided into several families, each of which is composed of different members (Juwono & Martinus, 2016; Craig, 2018; Pearl, Prodromou & Workman, 2008; Tang et al., 2005; Bolhassani & Rafati, 2008) (Table 3). HSPs can be internalized by APCs through receptor-mediated endocytosis and can also promote the activity of some cytokines,
Ser-Lys-Lys-Lys-Lys

R-NH-CH-COOH Cys     NH2-CH-CO-R     R-NH-CH-CO
CH2            CH2           CH2
S               S             S
Pam CH2        CH2          CH2

$CH_3(CH_2)_{14}CO-OCH$    $CH_3(CH_2)_{14}CO-OCH$    $CH_3(CH_2)_{14}CO-OCH_2$
$CH_3(CH_2)_{14}CO-OCH_2$    $CH_3(CH_2)_{14}CO-OCH_2$    $CH_3(CH_2)_{14}CO-OCH$

**Pam2Cys**: R=H       **MALP2**: R=GNNDESNISFKEK     **Pam2CSK4**:R=H
**Pam3Cys**: R=CH3(CH2)14CO    **FSL-1**: R=GDPKHPKSF      **Pam3CSK4**:R=CH3(CH2)14CO

(A)              (B)             (C)

**Figure 3 The chemical structures of different TLR2-targeting Pam lipopeptides.** (A) Pam2Cys and Pam3Cys lipopeptides. (B) MALP-2 and FSL-1 lipopeptides. (C) Pam2CSK4 and Pam3CSK4 lipopeptides.   

chemokines, and co-stimulatory molecules through the NF-κB signaling pathway, since HSPs are the molecular chaperones of the antigen epitopes in the APC-MHC I delivery pathway (*Robert, 2003*; *Zachova, Krupka & Raska, 2016*). HSPs affect the immune system in different ways because they can act as carriers of antigens, molecular chaperones, and ligands for related receptors (e.g., TLR4) (*Moyle, 2017*). Among the HSPs, HSP60 obtained from gram-negative bacteria has the ability to stimulate cells of both the innate and acquired immune system, functions as a linker between immune cells, and coordinates immunological activities. Therefore, HSP60 appears to be a promising potential component of subunit vaccines designed to provide protection from infections with gram-negative bacteria (*Bajzert et al., 2018*). In addition, some researchers have confirmed murine HSP110 (mHSP110) to be a biological adjuvant that significantly enhances the immune response of C57BL/6 mice to the $E7_{49-57}$ or $E7_{11-20}$ epitopes of h-2d restricted human papilloma virus (HPV) (*Ding et al., 2013*). In addition, the Gp96-peptide complex is considered to be a highly effective stimulator of MHC I-mediated antigen presentation; this strategy makes full use of the built-in adjuvant function and antigen transfer ability of Gp96 to induce cytotoxic immunity against widespread viral or tumor antigens (*Strbo et al., 2013*).

Mycobacterial HSP70 (mHSP70) is widely used as an intramolecular adjuvant for epitope-based vaccines, and the carboxyl terminal polypeptide binding area (HSP70 aa 359–610) of mHSP70 has a stimulating epitope that can combine with the CD40 receptor to stimulate the production of Th1-polarizing cytokines (e.g., IL-12, TNF-α, and NO) to induce DC maturation (*Wang et al., 2002*; *Suzue et al., 1997*; *Suzue & Young, 1996*). Compared with the T cell epitopes of other proteins, the HSP70 T cell epitope can be efficiently processed by APCs, so that the polypeptide binding region of HSP70 has higher affinity with MHC molecules (*Basu et al., 2001*; *Castellino et al., 2000*). The single HSP70, which does not fuse with other exogenous epitopes, only induces a very weak cellular and humoral immune response. Some studies have shown that removing the amino end of HSP70 (ATPase domain) and retaining only its carboxyl end (polypeptide binding region) as an antigen can produce a large amount of IL-12, TNF-α, NO, and chemokines (*Fu et al., 2013*). In contrast, neither the ATPase domain

**Table 3 Several major types of heat shock proteins.**

| HSP family/members | Intracellular localization | Biological function | Reference |
|---|---|---|---|
| HSP60/HSP58, HSP60, HSP65 | Mitochondrion cytoplasm | It plays a role in the folding of proteins in the mitochondrial matrix. Hsp60 can affect T cell response in two ways: as a ligand of toll-like receptor 2 signalling and as an antigen. | Juwono & Martinus (2016), Bajzert et al. (2018) |
| HSP70/HSP68, HSP70, HSP72, HSP73 | Cytoplasm or nucleus Mitochondrion Endoplasmic reticulum | It plays a role in different cell processes, from protein folding to protein complex decomposition and cell membrane protein transfer. Almost every protein that is not folded into its original state has multiple accessible Hsp70 binding sites. It is most commonly used as an adjuvant and protective antigen. | Craig (2018), Cheng et al. (2014b) |
| HSP90/HSP83, HSP84, HSP87, HSP90, Gp96 | Cytoplasm or nucleus Endoplasmic reticulum Golgiosome | HSP90 regulates the stability of client proteins, activates intracellular division of labor, participates in the regulation of multiple signaling pathways and cell cycle processes, and plays an important role in carcinogenic signal transduction, anti-apoptosis, metastasis, stress injury, autoimmune and other diseases treatment. HSP90 can promote the correct assembly, folding, or restoring the normal conformation of the damaged protein, prevent the wrong folding and aggregation of the protein and also promote the processing of MHC I antigen through the generation and assembly of the antigen determinant cluster of 26s protease complex. | Pearl, Prodromou & Workman (2008), Strbo et al. (2013) |
| HSP110 | Cytoplasm or nucleus | HSP110 has a strong molecular chaperone function and can present antigen peptides to APCs to activate specific antitumor cellular immunity. Moreover, HSP110 can also up-regulate the expression of MHC-II, CD40 and costimulatory molecules of APC, thus enhancing the antigen-presenting ability of APC. | Tang et al. (2005), Ding et al. (2013) |
| Small HSPs/HSP22 HSP23, HSP26, HSP27, HSP28, αβ-crystallin | Cytoplasm or nucleus Mitochondrion | Stable cytoskeleton Heme catabolism or antioxygenic property Actin dynamics | Bolhassani & Rafati (2008) |

**Note:**
The intracellular localizations and biological functions of several major types of heat shock proteins.

nor native HSP70 can induce such a powerful immune response (Cheng et al., 2014b; Li et al., 2006; Ge et al., 2006). For example, the fusion of the hantavirus glycoprotein (GP) and nucleocapsid protein with the carboxyl end of HSP70 can induce a more specific immune response (Cheng et al., 2014b). The major antigenic segment of the Japanese encephalitis virus E protein can fuse with the amino terminus of the peptide binding domain of HSP70, which can induce a more effective immune response

than the major antigenic segment of the E protein alone (*Ge et al., 2006*). HSP70s and HSP90s have been also found to act as carriers of tumor-derived peptides, adjuvants for antigen presentation, and can target the innate immune system by inducing anti-tumor immune responses (*Shevtsov & Multhoff, 2016*).

## Heparin-binding hemagglutinin

With regards to *Mycobacterium tuberculosis* antigens, several Mycobacterium lipid and glycolipid antigens (e.g., mycolic acid, lipoarabinomannan, glucose monomycolate) can be recognized by specific T cells by the CD1 antigen-presentation pathway in humans, suggesting the possible application of *Mycobacterium tuberculosis* lipid molecules in subunit vaccine preparation (*Moody et al., 2000*). *Dascher et al. (2003)* developed a vaccine which included lipids from *Mycobacterium tuberculosis* that were incorporated into liposomes with an adjuvant; the studies using a guinea pig aerosol tuberculosis challenge model demonstrated that lipid antigens play an important role in the immune response to tuberculosis infection, potentially through the production of CD1-restricted T cells. In addition, the *Mycobacterium tuberculosis* 6-kilodalton early secreted antigenic target protein (ESAT-6) is considered to be an important mediator in mycobacterial virulence, has strong antigenicity, and can induce a protective Th1 immune response against *Mycobacterium tuberculosis* (*Pandey et al., 2018*). *Khader et al. (2007)* vaccinated mice with an ESAT-6 peptide (amino acids 1–20 of ESAT-6) in an adjuvant composed of MPLA, trehalose dicorynomycolate, and dimethyl dioctadecylammonium bromide. The vaccination was found to induce antigen-specific T cells that produced IFN-$\gamma$, T cells that persisted in the central lymphoid organs, and antigen-specific IL-17-producing T cells that persisted in the lung (*Khader et al., 2007*). Recently, heparin-binding hemagglutinin (HBHA), a component of *Mycobacterium tuberculosis*, has been closely investigated for its strong immune potential, which can stimulate the migration of DCs and promote the expression of a variety of surface molecules (e.g., CD40, CD80, and CD86), MHC I and MHC II molecules, as well as inflammatory cytokines (e.g., IL-6, IL-12, IL-1$\beta$, and TNF-$\alpha$) in a TLR4-dependent manner (*Jung et al., 2011*; *Eraghi et al., 2017*). For example, HBHA can induce immunological protection against *Mycobacterium tuberculosis* by stimulating the production of IFN-$\gamma$, IL-2, and IL-17-coexpressing CD4[+] T cells (*Fukui et al., 2015*). As an effective immune adjuvant, HBHA can induce a strong Th1 cell immune response and plays an important role in the research of multi-epitope vaccines for immunotherapy (i.e., tumor vaccines). It has also been reported that the amino acid sequence of the epitope for the MAP1611 immunization region of the *Mycobacterium avium* subsp., paratuberculosis multiple antigenic peptide (MAP), is connected to the conservative amino acid sequence of HBHA. In addition, this recombinant subunit vaccine exhibited good immunogenicity and was identified to have a no allergenicity as predicted by employing a hybrid approach using the AlgPred program (*Rana, Rub & Akhter, 2014*; *Rana & Akhter, 2016*). Some researchers have designed a multi-epitope vaccine that includes CTL epitopes of Wilms tumor-1 and HPV E7 antigens, helper T lymphocyte epitopes of the tetanus toxin fragment C (TTFrC) and HLA PADRE, and HBHA as the intramolecular adjuvant

of epitope-based vaccines and is connected with appropriate linkers to enhance the effect of this recombinant multiple-epitope vaccine against cancer (*Nezafat et al., 2014*, *2015*). As a built-in adjuvant with a powerful immune enhancement effect, HBHA is rarely used in epitope vaccine research and is worthy of further study in the development of antiviral and cancer epitope-based vaccines.

## Bacterial flagellin

Flagellins are TLR5 and NOD receptor ligands that can activate both innate and acquired immune cells (*Huleatt et al., 2008*; *Wang et al., 2014*; *Hajam et al., 2017*). Both the N- and C-terminals of flagellin are composed of conservative alpha helices that function as TLR5 recognition sites, and the portion between the N- and C-terminals comprises a highly variable flagellin antigen region (*Murthy et al., 2004*). It has been found that deletion of part of the highly variable region of flagellin disables the ability of the host to produce antibodies against bacterial flagellum but does not affect its adjuvant activity (*Deng et al., 2017*). Therefore, researchers typically replace this region with exogenous antigen epitopes (e.g., the HPV prophylactic peptide vaccine) (*Nempont et al., 2008*; *Negahdaripour et al., 2017a*). For example, replacing the FliC variable region with the M2e protein of influenza A does not obstruct TLR signaling pathways (*Smith et al., 2003*; *Deng et al., 2017*). A truncated flagellin (tFL) with deletion of the hypervariable regions was used as a carrier by chemical conjugation with a malaria antigen M.RCAg-1 (M312), and compared with the physical mixture of M312 and tFL, the conjugates M312-PEG-tFL elicited higher M312-specific antibody titers (*Guo et al., 2018*). In addition, two HPV epitopes and some universal Th epitopes have been linked to the different flagellin positions via different linkers, and the optimal construction of a multiepitope vaccine was screened using protein structure analysis, modeling, optimization, and an evaluation of immunogenicity (*Negahdaripour et al., 2018*). In addition, four copies of the ectodomain of matrix protein 2 (f4M2e) of the influenza A virus (IAV), H1, HA2 domain (fHApr8), or H3 HA2 domain (fHAaichi) were used to replace the high immunogenicity region of flagellin, and the fusion proteins were crosslinked with propionate (DTSSP) to form protein NPs, thereby retaining the agonist activity of FliC to TLR5, and ability to assist the epitope protein in stimulating the immune response against IAV (*Deng et al., 2017*).
The combination of flagellin with multiple copies of HPV L2 neutralization epitopes have demonstrated a strong broad spectrum anti-HPV effect without the participation of other adjuvants, thus demonstrating a significant advantage of this strategy in enhancing the cross-protection of the HPV vaccine (*Kalnin et al., 2014*, *2017*; *Gambhira et al., 2007*). In addition, *Ajamian et al. (2018)* found that inserting the HIV gp41$_{607-683}$ (MPER) into a flagellin-based scaffold could significantly enhance the immunogenicity of gp41$_{607-683}$ in a TLR5-dependent manner and induce strong humoral responses specific to MPER in a mouse model. Various flagellin antigen fusion proteins have been studied in human clinical trials. Furthermore, flagellin is also commonly used as an antigen skeleton of SAPN-based vaccines (*El et al., 2017*), for which the associated content is described below.

## Outer membrane vesicles

Outer membrane vesicles (OMVs) are naturally secreted on the surface of most gram-negative bacteria, and the vesicle membrane typically consists of lipopolysaccharide (LPS), glycerophospholipids, and outer membrane proteins (OMPs) (*Tan et al., 2018*). Due to their intact outer membrane and periplasmic contents, OMVs possess good intrinsic stimulation ability and strong immunoreactivity, which can induce strong humoral and cellular immune responses. In addition to research focused on the possibility of using OMVs as candidate antigens for vaccine development, there is growing interest in the application of OMVs as a self-adjuvant for immunostimulatory molecules. This function is mediated by the interaction between the OMV-associated PAMPs and the TLRs expressed on the surface of APCs, thus enhancing the immune response to exogenous antigens (*Gnopo et al., 2017*). OMVs can also even be used as a mucosal transporter to transport antigens to the mucosal barriers (*Jang et al., 2004*). In addition, designed glycoengineered OMVs, which can display the O-antigen and surface glycans from different bacteria could be used as bacterial vaccine platforms to prevent bacterial infections (*Valguarnera & Feldman, 2017*). In addition, plasmids can be transported into OMVs to further modify the intracavity content, including LPS functionality and attenuate toxicity (*Tan et al., 2018*). *Hekmat et al. (2018)* developed a novel hepatitis C virus (HCV) therapeutic vaccine candidate, rC/N-NMB OMVs, formulated as a targeted synthesized recombinant fusion protein consisting of a truncated core and NS3 (rC/N) of HCV as a bipartite antigen accompanied by Neisseria meningitidis serogroup B OMVs (NMB OMVs), has the ability to induce Th1, Th2, and Th17 immune responses. Compared with MF59 and Freund adjuvant, NMB OMVs can significantly increase the level of Th1 immune responses (*Hekmat et al., 2018*). *Liu et al. (2018)* have demonstrated OMVs from flagellin-deficient *Salmonella Typhimurium* can serve as an adjuvant when combined with OMPs from different Salmonella serotypes, and enhances the cross-protection capacity of this combined vaccine. Additionally, novel vaccine adjuvant OMVs have been reported, which can serve as delivery carriers to improve vaccine safety and protective efficacy. ClyA is a 34 kDa pore-forming toxin enriched on OMVs, for which exogenous antigens are fused at the C terminus of ClyA to produce ClyA-antigen fusion proteins on OMVs (*Kim et al., 2008*). *Chen et al. (2010)* demonstrated that rOMVs carrying ClyA-GFP fusion proteins could induce a high level of anti-GFP IgG titers in mice, which was similar to that of GFP adjuvanted with alum. *Rappazzo et al. (2016)* immunized mice with ClyA-M2e4xHet OMVs displaying an influenza-derived antigen, M2e4xHet, which was associated with 100% survival following subsequent influenza challenge. In addition to ClyA, other prospective proteins also have the potential to display an antigen of interest, such as the hemoglobin protease (Hbp) autotransporter platform (*Daleke-Schermerhorn et al., 2014*). *Jong et al. (2014)* introduced a mutation to preserve the integrity of Hbp to avoid cleavage following translocation to the outer membrane. *Kuipers et al. (2015)* demonstrated that rOMVs displaying the pneumococcal antigens, pneumococcal surface protein A and pneumolysin, by the Hbp system could prevent pneumococcal colonization. Although

some obstacles to the development of OMV adjuvants remain (e.g., large number of clinical and preclinical assessments, and limited knowledge of the OMV manufacturing process), we believe that the use of OMVs as an epitope-based vaccine delivery system would also be of great value in controlling all types of pathogen infections due to their comprehensive immune potency, higher safety, and substantial mucosal delivery efficacy (*Tan et al., 2018*).

In addition, generalized modules for membrane antigens (GMMA) as an OMV technology are outer membrane particles consisting of outer membrane lipids, OMPs, and soluble periplasmic components (*Gerke et al., 2015*). GMMA are derived from gram-negative bacteria (i.e., Salmonella and Shigella) which are genetically modified (deletion of the tolR gene) to enhance the associated advantages of being cost-effective with high-production yields. Further gene deletions (i.e., the late acyltransferases genes HtrB175 and MsbB) resulted in GMMA with penta-acylated LPS with no possibility of infection (*Rossi et al., 2014*). Due to the self-adjuvanting properties of GMMA that deliver innate immune signaling through PAMPs (i.e., TLR ligands), many studies have shown that GMMA vaccines can simultaneously deliver surface polysaccharides and OMPs to the immune system and display greater immunogenicity compared to glycoconjugate vaccines (*Micoli et al., 2018*; *MacLennan, Martin & Micoli, 2014*). Moreover, it has been demonstrated that the GMMA technique can be used as a carrier to display the salmonella LPS O-antigen to the immune system. It is envisaged that GMMA could also be considered as a built-in adjuvant platform for epitope-based vaccines against pathogens other than gram-negative bacteria. More importantly, clinical trials (currently under way with *Shigella sonnei* GMMA vaccines) are required to further assess the safety and tolerance of this vaccine platform in humans.

## Salmonella porin

Salmonella Typhi expresses a variety of porins. While the major S. Typhi OmpC and OmpF porins can be expressed constitutively, the expression of other porins (e.g., OmpS1 and OmpS2) is relatively low in vitro and during potential infection (*Perez-Toledo et al., 2017*). Both the major and minor S. Typhi porins can effectively activate the innate immune system through the TLR2 and TLR4 signaling pathways, resulting in the increased expression of costimulatory molecules and cytokines in DCs and B cells (*Moreno-Eutimio et al., 2013*; *Cervantes-Barragan et al., 2009*). Due to such immune-activating properties, some of these porins have been used as potential vaccine adjuvants. *Perez-Toledo et al. (2017)* have shown that the S. Typhi porins, OmpC and OmpF, are multipurpose vaccine adjuvants, which can be used to polarize the T cell response toward a Th1/Th17 profile and enhance the antibody response generated toward T-dependent and T-independent antigens with poor immunogenicity. In addition, *Leclerc et al. (2013)* used malva mosaic virus NPs as a vaccine platform to improve the stability of the M2e peptide of influenza A in conjunction with OmpC purified from Salmonella typhi as an adjuvant; their data demonstrate that OmpC increased the immune response to the M2e peptide and provided protection against a heterosubtypic influenza strain in a mouse model. Moreover, *Moreno-Eutimio et al. (2013)* investigated the
immunogenic and protective capacities of the OmpS1 and OmpS2 porins and determined that these porins can be potent inducers of the innate immune response, exhibiting adjuvant properties that can promote increased antibody titers and long-term antibody responses when co-immunized with antigens.

### β-Defensin

β-Defensins are antimicrobial peptides involved in the innate immune response of the host and are responsible for stimulating innate and adaptive immune responses by recruiting naïve T cells and immature DCs through interactions with corresponding immune receptors (e.g., CCR6 or TLRs) (*Narula et al., 2018*). *Kim et al. (2018)* have concluded that human β-defensin 2 can induce the primary antiviral innate immune response and may also mediate the induction of antigen-specific immune response against a conjugated antigen. Using immunoinformatic methods, a multi-epitope vaccine for dengue was developed that included Tc and Th cell epitopes with β-defensin included as a molecular adjuvant at the N-terminal of the construct (*Ali et al., 2017*). Similarly, researchers developed an anti-chikungunya multi-epitope vaccine that included B cell and T cell-binding epitopes and IFN-γ inducing epitopes with β-defensin added as a built-in adjuvant (*Narula et al., 2018*).

## VIRUS-LIKE PARTICLES AS BUILT-IN ADJUVANT PLATFORMS

### Hepatitis B virus core antigen

Hepatitis B virus core antigen can act both as a Th cell-dependent or Th cell-independent antigen (*Roose et al., 2013*), and the Th-priming effects of HBcAg can easily transfer the adaptive response to the inserted related epitopes (*Milich et al., 1987*). In addition, the nanoscale structure of HBcAg can be more effectively identified and processed by APCs (*Lee et al., 2009*; *Ong, Tan & Ho, 2017*). Therefore, HBcAg has been used as an vaccine carrier for several exogenous pathogens (e.g., hepatitis B, C, and E virus, influenza virus, foot-and-mouth disease virus, Human enterovirus 71, coxsackievirus A16, and *C. trachomatis*), and the immunogenicity of recombinant HBc-based VLP vaccines against pathogens has also been verified in animal models (*Dai et al., 2016*; *Su et al., 2013*; *Zheng et al., 2016*; *Chu et al., 2016*; *Zhu et al., 2016*; *Wu et al., 2017*; *Jiang et al., 2017*). VLPs are the self-assembled structural proteins of most viruses and can stimulate the immune response in the absence of an adjuvant by exposing pathogen epitopes and simulating the structure of natural viruses (*Plummer & Manchester, 2011*; *Yang et al., 2016*). In addition, VLPs can stimulate an innate immune response by activating PRRs (e.g., TLRs) (*Shirbaghaee & Bolhassani, 2016*). In addition, the autoantigen molecules displayed by HBcAg VLPs can escape immune tolerance and produce specific auto-antibodies (*Long et al., 2014*). Due to these advantages, the HBc is often used as a powerful carrier protein and built-in adjuvant to display exogenous epitopes (*Chen et al., 2017*; *Jiang et al., 2017*; *Roose et al., 2013*; *Liang et al., 2018*). In general, researchers will insert pathogen epitopes into the HBcAg major immunogenic region (MIR; HBcAg aa 78–82 in the spike tip of HBV), which does not affect the self-assembly of the fusion

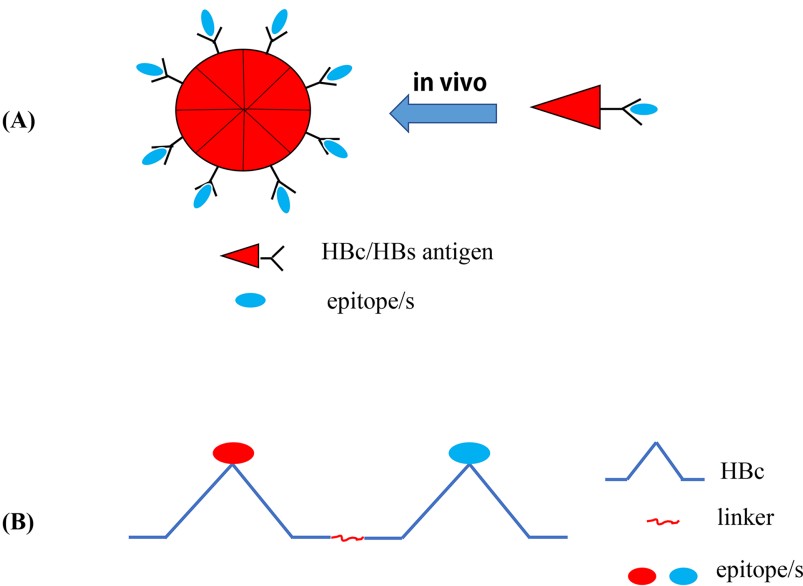

**Figure 4 Recombinant HBc-based VLPs or HBs-based VLPs.** (A) (1) The HBc proteins naturally form the dimers, the building blocks that forms the VLPs. It takes about 60 such dimers (i.e., 120 copies of HBc) to form a HBc-based VLP. The results showed that there were about 40 amino acid residues inserted into the N-terminal of HBc. In the MIR region of HBc, 50 or 100 amino acid residues can be inserted, and as many as 100 or more residues at the C-terminal do not interfere with the formation of particles. (2) Hepatitis B surface antigen (HBsAg) can also self-assemble into highly organized viroid particles with a diameter of 22 nm. These HBs-derived VLPs contain about 100 HBsAg molecules and provide a unique opportunity to display multiple exogenous epitopes. (B) Hepatitis B virus tandem core platform. The two replicas of HBc protein are linked together by covalent bonds through flexible amino acid sequences so that the fused dimers can be folded correctly and assembled into HBc particles. In the assembled HBc particles, the four helix bundles formed at each dimer interface appear on the surface as prominent "spikes". The tip of the spike is the preferred site for inserting foreign sequences for bivalent vaccine.

protein into VLP NPs, to generate immune activation; therefore, antigen epitopes will be presented on the surface of the particles (*Wang et al., 2017a*; *Chen et al., 2017*) (Fig. 4A). As the main insertion site of exogenous epitopes, MIR can significantly enhance the immunogenicity of such epitopes (*Dai et al., 2016*; *Reynolds et al., 2015*). Recently, *Wu et al. (2017)* developed a novel vaccine against chickenpox and hand-foot-mouth disease by constructing three VLPs with HBcAg used as a carrier protein, and epitopes derived from varicella-zoster virus-gE, EV71 (enterovirus71)-VP1, and EV71-VP2 were displayed by HBcAg. This study also fully demonstrated the significant potential of HBcAg as a carrier protein for epitope vaccines used in multivalent epitope vaccine research (*Wu et al., 2017*). The tandem core (TC) contains two HBcAg molecules that are connected by the appropriate linker and has two independent MIRs which can individually accommodate multiple exogenous antigens (Fig. 4B). This VLP platform is associated with beneficial prospects for the development of multivalent vaccines (*Roose et al., 2013*). Alejandro et al. inserted the four conserved antigenic regions of the matrix protein 2 ectodomain and hemagglutinin stalk of an existing IAV into the MIR region of TC, thereby constructing a VLP vaccine called Tandiflu1, which can induce the production of cross-protective antibodies (*Ramirez et al., 2018*).

## Hepatitis B virus surface antigen

The tertiary structure of hepatitis B virus surface antigen (HBsAg) can form a highly conserved hydrophilic loop. Additionally, it has been reported that the existing immunity to HBsAg does not impede the immune response to foreign epitopes carried by HBsAg particles, which is often used as a carrier to insert exogenous antigens into the external hydrophilic loop or the end of the HBsAg N- or C-terminal (*Bellier & Klatzmann, 2013*). *Czarnota et al. (2016)* inserted a highly conserved epitope (amino acid residues 412–425) of the HCV E2 GP into the hydrophobic loop of HBsAg, and the chimeric protein was then expressed in an unconventional *Leishmania tarentolae* expression system and independently assembled into VLPs, which demonstrated high immunogenicity and induced cross-reactive antibodies against HCV (*Czarnota et al., 2016*). *Wei et al. (2018)* also used HBsAg to display neutralizing HCV epitopes to obtain chimeric HCV–HBV VLPs as a novel strategy for developing a bivalent prophylactic HCV–HBV epitope vaccine (*Wei et al., 2018*). The envelope protein domain III (EDIII) of dengue viruses (DENV) contains good serotype-specificity and cross-reactive epitopes. *Ramasamy et al. (2018)* fused the EDIII of all four DENV serotypes with four copies of HBsAg and expressed the construct in the *P. pastoris* GS115 strain to obtain a tetravalent VLP vaccine termed DSV4, which has a high immunogenicity and produces effective and persistent neutralizing antibodies against all four DENV serotypes in mice.

## Phage-based VLPs

The over-expression of the single-chain dimer coat proteins (CPs) of MS2 and PP7 RNA phages in bacteria can spontaneously assemble into recombinant MS2 or PP7 VLPs containing 90 dimer copies and 90 exogenous epitopes. In addition, for the Qβ bacteriophage, 180 copies of single-chain dimer capsid proteins can spontaneously assemble into a VLP; thus, at least 180 exogenous epitopes can be displayed on the surface of a Qβ VLP (*Basu et al., 2018*). Recent reports have shown that the Qβ bacteriophage contains TLR ligands that can enhance Qβ-VLP-induced T cell-independent and -dependent Ab reactions, including a germinal center reaction via of TLR/MyD88 signaling in B cells (*Tian et al., 2018*). This VLP platform which contains no viral genome can be used in the future as a carrier system for the administration of safe vaccines against many pathogens (*Pumpens et al., 2016*). *Basu et al. (2018)* described potential B cell epitopes located on the envelope proteins of Zika virus on the surface of a highly immunogenic bacteriophage VLP platform (MS2, PP7, and Qβ), and evaluated the immunogenicity of these VLPs in mice. *Zhai et al. (2017)* also displayed consensus peptides from HPV L2 and tandem HPV31/16L2 peptides on the surface of bacteriophage MS2 VLPs. These MS2-L2 VLPs can induce high antibody titers in mice and are cost-effective vaccine candidates against HPV; however, HPV vaccines with greater cross-protection should be further evaluated to prevent more types of HPV (*Zhai et al., 2017*). Recently, Qβ VLPs have been applied as carriers for the development of carbohydrate-based anticancer vaccines (*Sungsuwan, Wu & Huang, 2017*). Additionally, researchers have developed a size-exclusion chromatography-based purification method for an VLP-based influenza A vaccine derived from the MS2 phage that displays an

epitope from the extracellular domain of the IAV matrix two protein. Moreover, the purification procedure provides an improved strategy for the future large-scale production of VLP-based epitope vaccines (*Lagoutte et al., 2016*).

## Tobacco mosaic virus

Tobacco mosaic virus (TMV) is a widely studied and identified filamentous plant virus. TMV particles are hollow with tubular rods (300 nm length × 18 nm diameter) consisting of about 2,130 CP subunits encase a single-stranded, plus-sense RNA genome (*Culver, 2002*). As an antigen carrier, TMV has two important functions: (1) due to the architecture and size of TMV, TMV carrying antigen epitopes is robustly and readily taken up by DCs, leading to the activation of key surface markers (*Smith et al., 2007*); and (2) TMV also provides adjuvant effects, due to either repetitive antigen displayed on the surfaces of TMV or the presence of non-functional viral RNA that is important for inducing cellular-mediated immunity (*Banik et al., 2015*). *Kemnade et al. (2014)* demonstrated that TMV is capable of boosting TMV-induced antigen-specific T cell responses, but does not induce neutralizing self-immunity. In addition, since TMV is not a human pathogen, it is intrinsically secure (*Liu et al., 2013*). These findings further confirm that TMV has great potential as an epitope-based vaccine vector. *McCormick et al. (2006)* fused well-characterized T cell epitopes that provide protection against tumor challenge in mice into a TMV CP and demonstrated that C57BL/6 mice inoculated with TMV displayed significantly improved protection against tumor challenge in both the EG.7-Ova and B16 melanoma models. Moreover, *Zhao et al. (2015)* reported that when efficient copper (I)-catalyzed azide–alkyne cycloaddition reaction was performed for the conjugation of the small molecule estriol (E3) onto TMV capsid, TMV can induce a strong and long-term antibody response. Furthermore, *Banik et al. (2015)* developed a multivalent subunit vaccine against tularemia using a TMV-based delivery platform and demonstrated TMV can serve as a suitable built-in adjuvant for multiple protective antigens (PAs) of *F. tularensis*, as well as induce cell-mediated immune responses and long-lasting humoral immunity against tularemia.

## Papaya mosaic virus

As a member of the potexvirus family, papaya mosaic virus (PapMV) displays a flexible rod-like structure composed of 1,400 subunits of the viral CP assembled around a positive-strand RNA (*Lacasse et al., 2008*). Since PapMV-based VLPs comprised of PapMV CPs assembled around an ssRNA can efficiently trigger an innate immune response, they can be used as a vaccine adjuvant platform. Following phagocytosis, PapMV-based VLPs can reach the endosome of immune cells and release ssRNA, which engages and activates TLR7 (*Therien et al., 2017*). The direct fusion of antigenic peptides to the open reading frame (ORF) of the PapMV CP can lead to the formation of chimeric VLPs that can trigger a humoral or CTL response against the fused antigen (*Bolduc et al., 2018*). Additionally, *Carignan et al. (2015)* have shown that the fusion of a short M2e (sM2e) epitope (nine amino acids) to the N-terminus of the PapMV CP allows for the assembly of highly immunogenic VLPs. This group further demonstrated

that an intramuscular injection of PapMV-sM2e VLPs is sufficient to induce a powerful anti-M2e humoral response that protects mice against subsequent challenge with influenza A. Similarly, *Bolduc et al. (2018)* have developed a PapMV-based VLP vaccine candidate capable of inducing robust and broad protection against two different influenza A strains (H1N1 and H3N2). *Lacasse et al. (2008)* have shown that PapMV VLPs carrying the H-2b-restricted dominant p33 CTL epitope from the lymphocytic choriomeningitis virus can induce DC maturation and cross-presentation of the p33 CTL epitope, which triggers a protective antiviral T cell response. Furthermore, *Therien et al. (2017)* engineered a PapMV-based VLP platform with a SrtA receptor motif and allowed SrtA to attach to the long peptides of the VLPs. This approach was found to be more versatile than the fusion of only small peptides to the ORF of the PapMV CP. Therefore, PapMV NPs with SrtA-conjugated peptide antigens may represent a promising tool in vaccine design against various diseases (*Therien et al., 2017*).

## KEYHOLE LIMPET HEMOCYANIN AND BOVINE SERUM ALBUMIN

Keyhole limpet hemocyanin (KLH) and bovine serum albumin (BSA) are easily identified by the immune system as non-self components, which is useful for enhancing the immunogenicity of small antigens or a low antigen dose. KLH and BSA are easily processed APCs and can recruit Th cells to assist in antigen uptake (*Mora et al., 2017*).

These characteristics of KLH and BSA have promoted their frequent use as epitope carrier proteins. For example, researchers have developed the novel epitope peptide vaccine, Aβ3-10-KLH, by coupling the B cell epitope, Aβ3-10, from amyloid-β peptide (Aβ) with KLH, for the potential treatment and prevention of Alzheimer's disease (AD) (*Ding et al., 2016*, *2017a*). In another study, an anti-PCSK9 (proprotein convertase subtilisin/kexin type 9) peptide vaccine using KLH as the carrier protein was shown to produce long-lasting anti-PCSK9 antibodies and is considered to be the primary vaccine for the treatment of dyslipidemia in the future (*Kawakami et al., 2018*). In addition, a short peptide (UPK3A 65-84) from Uroplakin 3A (UPK3A) covalently coupled with KLH and CpG as adjuvant was found to be an immunotherapeutic vaccine for bladder cancer (*Izgi et al., 2015*). In addition, BSA is often used as a carrier protein for small antigens in glycoconjugate vaccines (*Prasanphanich et al., 2015*). For example, *Cai et al. (2013)* combined a synthesized MUC1 glycopeptide with BSA or three different T-helper cell epitopes of TTox and demonstrated a beneficial effect. Furthermore, the immune complex formed by coupling the synthetic trisaccharide Galα(1,3)Galβ(1,4)GlcNAcα of *Trypanosoma cruzi* with BSA as a carrier protein was reported to be a vaccine candidate for Chagas disease (*Schocker et al., 2016*).

## BACTERIAL TOXIN PROTEINS

### Heat-labile toxins and cholera toxin

Heat-labile toxins (HLTs) are produced by some enterotoxigenic *E. coli* strains and can be fused with other antigenic proteins to function as an adjuvant (*Da et al., 2011*; *Luiz et al., 2015*; *Hu et al., 2014*) (Table 4). HLTs and cholera toxins (CTs) are highly homologous, consisting of five subunit-Bs and one subunit-A, and are members of

**Table 4 Summary of several bacterial toxin build-in adjuvants listed in this paper.**

| Objective | Advantages and characteristics | Application example | Reference |
|---|---|---|---|
| Heat labile toxins (HLT) | 1. B subunit of LT or the mutant form of LT can activate the dendritic cells and B and T lymphocytes. | 1. Fused the Heat-labile LTB with the linear B cell epitope of Aeromonas hydrophila outer membrane protein (OmpC) or two epitopes of Zairian Ebola virus GP1 protein. | *Rodrigues et al. (2011)*, *Sharma et al. (2017)*, *Rios-Huerta et al. (2017)* |
| Cholera toxin (CT) | 1. CTB's strong affinity to GM1 ganglioside receptor.<br>2. Reduce the minimum concentration of antigens required for activation of immune cells. | 1. A multivalent epitope-based vaccine CWAE against h. pylori and anti-atherosclerosis multi-epitope vaccine.<br>2. CTB-Human Mucin 1(MUC1) vaccine. | *Guo et al. (2017)*, *Tourani, Karkhah & Najafi (2017)* |
| Diphtheria toxin (DT) | 1. CRM197 is a mutant of DT, which can effectively combine and present peptides and rapidly activate CD4 T cells by multiplicity of Th1 and Th2 cytokines.<br>2. The DTT is no safety hazard and contains four Th cell epitopes.<br>3. DTT can form a turn-helix-turn structure completely exposed to the surface, which may be a potential site for insertion of exogenous epitopes. | 1. Several short B cell epitopes on the Her-2/neu protein were coupled with CRM197.<br>2. The epitope of TNF-α is coupled to the insertion site of DTT, developed an anti TNF-α vaccine DTNF. | *Tobias et al. (2017)*, *Zhang et al. (2016)* |
| Tetanus toxoid (TT) | 1. TT has multiple CD4+ Th cell epitopes and associated memory Th subsets.<br>2. Helper epitopes selected from Tetanus toxin fragment C (TTFrC). | 1. A new type of anti-gastrin vaccine.<br>2. As the carrier protein of glycoconjugate vaccine.<br>3. The anti-brucellosis multi-epitope vaccine and anti-atherosclerosis multi-epitope vaccine. | *Saadi, Karkhah & Nouri (2017)*, *Broker (2016)*, *Arcuri et al. (2017)* |
| Anthrax toxin | 1. The N-terminal (the first 255 amino acids) of lethal factor (LF) of anthrax toxin termed LFn, retains protective antigen (PA)-binding and translocation capabilities but has no toxic activity.<br>2. LFn has been used to transfer foreign proteins and peptides into the cytoplasm. | 1. A chicken ovalbumin (Ova) recombinant protein (LFn-Ova).<br>2. LFn as the delivery carrier of ESAT-6 antigen. | *Wesche et al. (1998)*, *Shaw & Starnbach (2008)*, *Chandra et al. (2006)* |

**Note:**

The advantages and characteristics and some application examples of several bacterial toxin build-in adjuvants.

the bacterial protein toxin AB5 family. Subunit-A is a toxic subunit, noncovalently bound to the B pentamer and has ADP ribosyltransferase activity, whereas subunit-B is a nontoxic receptor-binding subunit (*Lencer, Hirst & Holmes, 1999*). Generally, non-toxic HLTs (e.g., subunit-B and mutant forms of HLT) also act as immune adjuvants to activate DCs, B cells, and T cells, regulate epitope specificity, and improve the immune response (*Batista et al., 2014*; *Rodrigues et al., 2011*). The fusion of the HLT B subunit (HLTB) with the linear B cell epitope of the outer membrane protein (OmpC) of *Aeromonas hydrophila* can stimulate the production of neutralizing antibodies against this linear epitope and generate a Th2 type mixed auxiliary T cell immune response

(*Sharma et al., 2017*). The two epitopes of the Zairian Ebola virus GP1 protein, which can be recognized by neutralizing antibodies, were coupled with HLTB protein to form recombinant antigen HLTB-EBOV expressed in plant tissues, and immunizing mice with the recombinant antigen presented by the plant induced a higher level of IgA and IgG responses (*Rios-Huerta et al., 2017*). The CT subunit-B (CTB) can be used as a powerful adjuvant to generate mucosal immunity due to its strong affinity to the GM1 ganglioside receptor which is primarily located on mucosal epithelial cells (i.e., M cells) (*Pinkhasov et al., 2010*). The multivalent epitope-based vaccine against *Helicobacter pylori*, CWAE, and an anti-atherosclerosis multi-epitope vaccine have been developed using CTB as intramolecular adjuvant (*Tourani, Karkhah & Najafi, 2017*; *Guo et al., 2017*).

## Diphtheria toxin

The carrier protein cross-reacting material 197 (CRM197) is an inactivated and non-toxic form of diphtheria toxin (DTT) created using an enzymatic reaction, and has been successfully applied in many vaccines against infectious diseases because it can effectively combine and present peptides (*Caro-Aguilar et al., 2013*) (Table 4). Moreover, CRM can rapidly activate CD4$^+$ T cells by generating a multitude of Th1 and Th2 cytokines, thereby promoting the proliferation of B cells and regulating the level of antibody production (*Kamboj et al., 2001*). Several short B cell epitopes (P4, P6, and P7) on the Her-2/neu protein were combined with the polyepitope peptide, P467, using CRM197 as a carrier protein to conjugate with this complex epitope, demonstrated a strong anti-tumor response (*Tobias et al., 2017*). The function of DTT is to assist the enzyme active region (C-domain) in passing through the endocytosis membrane, and there are no associated risks when the transmembrane domain of DTT is used as the protein carrier for exogenous antigen (*Ladokhin, 2013*; *Malito et al., 2012*). *Xu et al. (2017)* developed a vascular endothelial growth factor (VEGF)-based antigen DTT-VEGF consisting of the receptor-binding domain of VEGF and DTT stimulated neutralizing antibody response and induced type 1 immune response as well as anti-tumor CTLs in mice, and their data demonstrated that DTT is an effective antigen carrier to break immune self-tolerance and DTT-VEGF has potential to be used a promising cancer vaccine. In addition, DTT contains four Th cell epitopes (aa 69–88, 119–138, 129–148, and 149–168) and the 89–96 amino acid residues form a turn-helix-turn structure that is completely exposed to the surface, which may be a potential site for the insertion of exogenous epitopes (*Diethelm-Okita et al., 2000*). For individuals who have been previously vaccinated with the DTT vaccine, the Th cell epitopes based on the DTT vaccine will induce a rapid CD4$^+$ memory T cell response (*Fraser et al., 2014*). For example, a TNF-α epitope has been coupled to the insertion site of DTT to develop an anti-TNF-α vaccine, DTNF, demonstrating the potential advantage of a DTT-based epitope vaccine gainst autoimmune diseases (*Zhang et al., 2016*).

## Tetanus toxoid

Since tetanus toxoid (TT) is a carrier protein possessing multiple CD4$^+$ Th cell epitopes and is associated memory Th subsets, it can be recognized by APCs and presented to

CD4+ Th cells (Table 4). As mentioned previously, these Th cells can provide the second signal required for B cell activation (*Van Der Heiden et al., 2017*; *Da et al., 2017*). Recently, it has been reported that a new type of anti-gastrin vaccine using TT as a carrier protein for multiple complex antigens can significantly enhance the immunogenicity of the vaccine (*He et al., 2018*). *Jarzab et al. (2018)* used TT as the carrier protein for the several synthetic linear or cyclic OmpC epitope peptides and demonstrated that cyclic peptide conjugated to TT as a potential candidate gainst shigellosis. Helper epitopes selected from TTFrC are typically associated with the target epitope to stimulate a CD4+ T cell response (e.g., anti-brucellosis and anti-atherosclerosis multi-epitope vaccines) (*Saadi, Karkhah & Nouri, 2017*; *Tourani, Karkhah & Najafi, 2017*). TT is also commonly used as a carrier protein for glycoconjugate vaccines (*Broker, 2016*). For example, the Vi polysaccharide of typhoid has been combined with TT via chemical bonding to compare the immunogenicity with that of diphtheria toxoid (DT) and CRM197 as a carrier protein (*Arcuri et al., 2017*).

### Anthrax toxin

The lethal toxin produced by *Bacillus anthracis* is a bipartite toxin consisting of PA as the cell binding moiety and lethal factor (LF) as the effector component. PA has the inherent ability to transport the enzymatically active LF across the host cell membrane into the cytoplasm, leading to the death of the host cell (*Liu et al., 2017*). PA binds to receptors on host cells, and the resulting PA heptopolymer can bind to three LF molecules. Subsequently, the entire toxin complex is endocytosed by cells. PA undergoes conformational changes due to endocytosis-associated acidification, resulting in transmembrane pores which can facilitate the translocation of LF molecules from the endosome into the cytosol (*Shaw & Starnbach, 2008*; *Arora, Misra & Sajid, 2017*). The N-terminal (the first 255 amino acids) of LF (PA binding region) termed LFn, retains PA-binding and translocation capabilities but has no toxic activity. LFn has been used to transfer foreign proteins and peptides into the cytoplasm, where they are processed through the MHC class I antigen presentation pathway, and subsequently induce CTL responses. However, it has been reported that LFn (as a fusion protein) cannot transfer all proteins into the cytoplasm (*Wesche et al., 1998*). *Shaw & Starnbach (2008)* fused two epitopes (one CD4+ T-cell epitope and one restricted epitope by MHC-I) from chicken ovalbumin (Ova) to LFn and demonstrated that this recombinant protein induced both Ova-specific CD4+ T cell and Ova-specific CD8+ T cell responses in mice. Additionally, *Chandra et al. (2006)* demonstrated that the anthrax toxin system can be used as an ESAT-6 delivery carrier of to induce CTL response against tuberculosis by the ability of LFn to deliver genetically fused ESAT-6 into the cytosol.

## OTHER POTENTIAL CO-DELIVERY SYSTEMS OF EPITOPE-BASED VACCINES

### Multiple antigenic peptide

Although the synthesis of long linear peptides with one or more epitopes can promote their presentation on MHC I and II molecules and enhance their immunogenicity, these vaccines continue to fail to demonstrate adequate efficacy or improve the overall survival
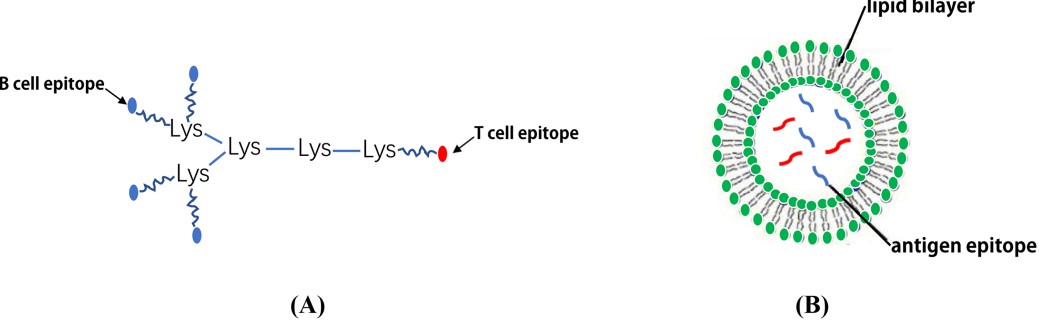

**Figure 5 The schematic diagram of MAP system and LCP nanoparticles.** (A) MAP epitope vaccine based on lysine scaffold. (B) The LCP nanoparticles.

rate (*Simanovich et al., 2017*). To solve this problem, epitopes can be displayed on a MAP system which contains a core matrix of lysine residues that form a scaffold (*Tam, 1988*). Currently, the most common strategy is to couple a number of epitope peptides to the dendritic polylysine-scaffold using standard solid phase chemistry (*Moyle et al., 2006*; *Horvath et al., 2004*) (Fig. 5A). It has been reported that the MAP-based vaccine, (B4T(thi)), which is composed of four copies of B cell epitopes (amino acid (aa) residues 136–154 of the FMDV VP1 protein) which are linked to a T cell epitope (aa residues 21–35 of the FMDV non-structural protein 3A) via thioether bonds can significantly induce an immune response against FMDV (*Cubillos et al., 2012*). Moreover, *Wen et al. (2016)* designed a novel tetra-branched MAP vaccine, M2e-MAP, which combines four copies of M2e with a foreign Th epitope to provide cross-protection against influenza viruses and may serve as a promising platform for influenza vaccine development. The immunogenicity of both adjuvanted and non-adjuvanted MAP vaccines composed of three conserved HCV envelope peptides (E1 peptide (aa 315–323) and E2 peptide (aa 412–419 and aa 516–531)) were studied. The results showed that the three HCV envelope MAP peptides exhibit strong immunogenicity and produce higher levels of neutralizing antibodies (*Abdelhafez et al., 2017*). Tumor vaccines based on MAP may also be an effective way to treat and prevent certain types of cancer (*Simanovich et al., 2017*).

## Self-assembled peptide nanoparticles

In the field of epitope vaccine research, natural self-assembled particles that are often used by researchers are proteins derived from viruses (HBsAg or HBcAg (mentioned above) and (TMV) capsid proteins) (*Lopez-Sagaseta et al., 2016*). With the development of advanced molecular machinery and the construction of sophisticated instruments and materials at an atomic level, a wide-range of materials are being used in SAPN systems (*Yang et al., 2012*). The SAPN complex is primarily dependent on the selection of suitable building blocks. In the SAPN β-sheets, both polar and hydrophobic amino acids are arranged in an alternating pattern, and self-assembly can occur spontaneously under suitable conditions (*Mandal, Nasrolahi & Parang, 2014*) (Fig. 6). *Indelicato, Burkhard & Twarock (2017)* designed a mathematical procedure for the structural
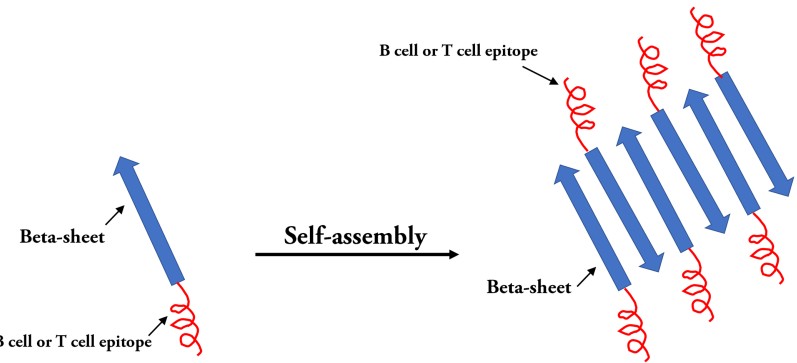

**Figure 6 Self-assembled peptides nanoparticles (SAPNs).** Systematic self-assembling peptides (β-sheet nanofiber vaccine) with antigen epitopes. 

classification of a specific class of SAPNs, which provides a toolkit for a systematic exploitation of SAPNs and predicting the density of epitopes on the SAPN surface. In addition, epitopes can be integrated on the surface of peptide NPs by synthesizing in the SAPN system (*Friedrich, Beasley & Rudra, 2016*). In addition, monomeric peptides containing self-assembled regions and epitopes can be produced by high throughput expression in vitro (*El et al., 2014*). Recent studies have shown that the SAPN malaria vaccine, FMP014, based on flagellin can be produced by replacing the D0 and D1 regions of bacterial flagellin with repeated sequences of several antigenic epitope regions (e.g., αTSR and circumsporozoite protein (PfCSP) of *Plasmodium falciparum*) (*Seth et al., 2017*). In addition, a vaccine was reported to be constructed by combining two conserved influenza virus antigens, M2e and Helix C, with SAPNs as the carrier and flagellin as the self-assembled adjuvant, and the experimental results from animal models show that the SAPN vaccine demonstrates substantial potential for the prevention and control of influenza viruses (*Karch et al., 2017*). A novel peptide-based SAPN HPV16 vaccine may also be a promising method of improving the efficacy of cervical cancer vaccines and can be used as a useful reference for the study of virus-related diseases and specific tumor immunotherapy (*Tang et al., 2012*).

## Lipid core peptide

As a form of vaccine delivery, the lipid core peptide (LCP) system allows for lipid amino acids in water with a poly-lysine core to combine with exogenous epitopes to form NPs displaying a polydispersity of 0.3–0.5 (*Skwarczynski & Toth, 2011b*). It appears that the key to self-assembly into small NPs is to ensure there is a proper balance between the composition of hydrophilic and hydrophobic components (*Schulze et al., 2017*). In the process of vaccine development, antigen epitopes are surrounded by phospholipid bimolecular layers, thus avoiding the degradation of antigenic peptides by enzymes (*Azmi et al., 2014*) (Fig. 5B). Using the LCP vector system, the B cell epitope, J14, on the M protein of *Streptococcus pyogenes* and the epitope peptides of the SfbI protein can be coupled (*Zaman et al., 2012, 2011*; *Moyle et al., 2014*). One study demonstrated that an LCP system using BPPCysMPEG as a mucosal adjuvant was more effective at

**Table 5 Different investigational built-in adjuvants for epitope-based vaccines.**

| Build-in adjuvant | Disease | Clinical phase | Reference |
|---|---|---|---|
| Gp96 | Late stage melanoma | Pilot | *Shevtsov & Multhoff (2016)* |
| | Metastatic colon carcinoma | Phase I | *Mazzaferro et al. (2003)* |
| | Gastric carcinoma | Phase I | *Shevtsov & Multhoff (2016)* |
| | Pancreatic carcinoma | Phase I | *Maki et al. (2007)* |
| | Hodgkin lymphoma | Phase I | *Shevtsov & Multhoff (2016)* |
| | Glioblastoma | Phase I–II | *Bloch & Parsa (2014)* |
| HSP70 | Malignant melanoma | Phase I | *Shevtsov & Multhoff (2016)* |
| | Chronic lymphatic leukemia | Phase I | *Shevtsov & Multhoff (2016)* |
| | Advanced solid tumors | Pilot | *Guzhova et al. (2013)* |
| | Glioblastoma | Phase I | *Guzhova et al. (2013)* |
| | HIV | Phase I | *SenGupta et al. (2004)* |
| Bacterial flagellin | Bacterial diarrhea and Guillain–Barré syndrome | Phase I | *Moyle (2017)* |
| | Influenza A virus | Phase I/II | *Taylor et al. (2011, 2012)* |
| | Dengue viruses/Zika virus | Preclinical | *Liu et al. (2015)* |
| | Respiratory syncytial virus | Preclinical | *Liu et al. (2015)* |
| MALP-2 | Pancreatic cancer | Phase I/II | *Schmidt et al. (2007)* |
| HBcAg | *P. falciparum* | Phase I | *Roose et al. (2013)* |
| | Influenza A virus | Phase I | *Roose et al. (2013)* |
| | Hepatitis B virus | Licensed | *Effio & Hubbuch (2015)* |
| Qβ VLP | Melanoma | Phase I, II, IIa | *Goldinger et al. (2012)* |
| | Persistent allergic asthma | Phase II | *Beeh et al. (2013)* |
| | Hypertension | Phase I | *Ambühl et al. (2007)* |
| | Nicotine dependence | Phase I | *Cornuz et al. (2008)* |
| | Alzheimer's disease | Phase I/IIa | *Winblad et al. (2012)* |
| SAPNs | Hepatitis B | Phase III | *Shirbaghaee & Bolhassani (2016)* |
| | Cervix cancer | Phase III | *Shirbaghaee & Bolhassani (2016)* |
| | Parvovirus porcine infection | Phase I/II | *Kushnir, Streatfield & Yusibov (2012)* |
| | Influenza A | Phase I/II | *Kushnir, Streatfield & Yusibov (2012)* |
| | Malaria | Phase III | *Kushnir, Streatfield & Yusibov (2012)* |
| | Alzheimer's disease | Phase II | *Lopez-Sagaseta et al. (2016)* |
| | Malignant melanoma | Phase II | *Lopez-Sagaseta et al. (2016)* |
| Liposome | Influenza | Phase I/II | *Tandrup et al. (2016)* |
| | Streptococcus mutans | Phase I | *Tandrup et al. (2016)* |
| | Neisseria meningitides | Phase I | *Tandrup et al. (2016)* |
| | HIV | Phase I | *Tandrup et al. (2016)* |
| | Mycobacterium tuberculosis | Phase I | *Tandrup et al. (2016)* |

| Build-in adjuvant | Disease | Clinical phase | Reference |
|---|---|---|---|
| PLGA | Hepatitis B | Clinical trial | Yang et al. (2016) |
| | HIV | Phase I | Bolhassani et al. (2014) |
| | Solid tumors | Preclinical | Bolhassani et al. (2014) |
| | Cervix cancer | Phase II/III | Bolhassani et al. (2014) |
| | Hepatitis C | Preclinical | Bolhassani et al. (2014) |
| Chitosan | RSV | Preclinical | Bolhassani et al. (2014) |
| | Tuberculosis | Preclinical | Bolhassani et al. (2014) |
| | Allergy | Preclinical | Bolhassani et al. (2014) |
| Gold nanoparticle | Influenza | Clinical trial | Vartak & Sucheck (2016), Zhao et al. (2014) |
| | HIV | Clinical trial | Vartak & Sucheck (2016), Zhao et al. (2014) |
| | RSV | Clinical trial | Vartak & Sucheck (2016), Zhao et al. (2014) |
| | Foot-and-mouth disease | Clinical trial | Vartak & Sucheck (2016), Zhao et al. (2014) |
| | Malaria | Clinical trial | Vartak & Sucheck (2016), Zhao et al. (2014) |

**Note:**
The clinical phases of various built-in adjuvants and their applications in the treatment of different diseases.

presenting synthetic epitope peptides (*Schulze et al., 2017*; *Olive et al., 2007*). Noushin et al. combined the antigen site of pre-fusion respiratory syncytial virus (RSV) F GP (Ø and II (B cell epitopes)) with PADRE (T helper cell epitope) using the LCP delivery system and found that the LCP constructs could induce a high level of RSV-specific antibodies (*Jaberolansar et al., 2017*). In addition, Nirmal et al. coated the group A streptococcus (GAS) lipopeptide-based vaccine candidate (LCP-1) on the surface of PLGA to form NPs that induced a high antibody response, suggesting that the PLGA-based LCP delivery system may be a promising method in vaccine research (*Marasini et al., 2016*). Due to several advantageous properties, lipid-based antigen complexes can effectively stimulate both humoral and cellular immune responses. Therefore, lipid-based delivery systems represent potential efficient vaccine adjuvants (*Kabiri et al., 2018*). For example, several liposome vaccines are currently being investigated in clinical studies (Table 5).

## Polymeric and inorganic nanoparticles

Some polymers exhibit good stability and biocompatibility, and can encapsulate and carry antigens to target cells (*Negahdaripour et al., 2017b*), such as PLGA, thermo-responsive synthetic polymers, and N-(2-hydroxypropyl) methacrylamide (HPMA) (*Li, Zhou & Huang, 2017*). Thus, these polymers are often used as organic biological carriers to present multiple epitopes to the immune system (*Tam, 1988*). Multi-alkyne-functionalized hbPG is a globular polymer with multiple branches that displays good biocompatibility and is not immunogenic; its multi-functional dendrimer-like structure provides sufficient

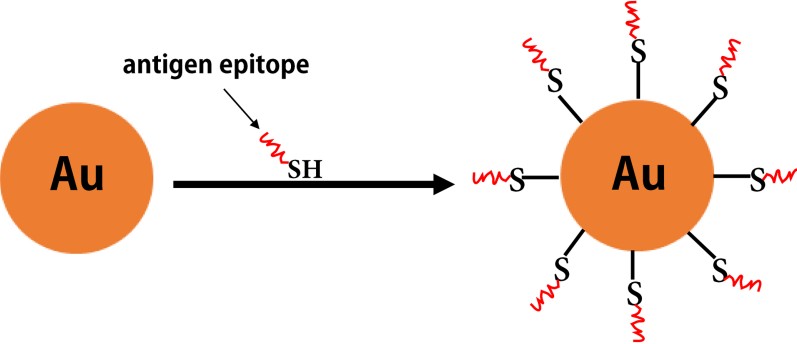

**Figure 7 The inorganic nanoparticles.** The formation of gold nanoparticles carrying antigen epitopes.

space to present multivalent antigens (*Glaffig et al., 2014*). Based on these characteristics, B cell epitopes of the tumor-associated antigen, MUC1, glycopeptide, and the T cell epitopes of the tetanus toxin, P2, can be coupled in series to each branch of the hbPG; moreover, these nanoscale branched spheres can express the glycopeptide on its surface, resulting in enhanced exposure of the antigens to the immune system (*Glaffig et al., 2014*, *2015*). Currently, various vaccines based on polymer NPs are being tested in preclinical and clinical trials, including those for tuberculosis, cancer, and HIV (Table 5).

Nanoparticles based on the conjugation of peptide epitopes and polymers display highly promising self-adjuvant properties; however, their lack of biodegradability may lead to serious deficiencies (*Skwarczynski et al., 2010*). Chitosan is a non-toxic, muco-adhesive, and biodegradable natural polymer that can be recognized by a variety of receptors (e.g., mannose receptors, TLR2, C-type lectin receptor, Dectin-1, and leukotriene B4 receptors) on the surface of APCs (*Li et al., 2013*). Chitosan NPs are typically prepared by interacting with anionic crosslinkers, antigens or polymers; anionic tripoly-phosphate was typically used as a crosslinking agent in previous studies (*Prego et al., 2010*). However, *Nevagi et al. (2018)* developed a novel chitosan NPs-based vaccine delivery system produced by the conjugation of a short anionic polymer (PGA) to a peptide antigen possessing a conserved B cell epitope derived from GAS and a universal Th epitope to form NPs with trimethyl chitosan via ionic interactions. The GAS peptide antigen-based chitosan NPs were formulated without the use of a crosslinking agent and evaluated in mice upon intranasal administration; such studies have determined that NPs can induce specific mucosal and systemic opsonic antibodies (*Nevagi et al., 2018*).

Some inorganic materials (e.g., gold, aluminum hydroxide, and carbon nanotubes) also have excellent biological properties (e.g., good biocompatibility, as well as ease of modification and processing). Among these, gold NPs have attracted increased attention as antigen carriers in vaccine research (*Negahdaripour et al., 2017b*) (Fig. 7). Compared with other inorganic NPs, gold NPs have been widely investigated in clinical studies for epitope-based vaccine delivery (e.g., influenza, HIV, and Malaria) (Table 5). For example, the Cap protein of porcine ring type 2 is used to directly react with AuNPs via a unique cysteine sulfhydryl group to form Cap-AuNPs, which expose neutralized

epitopes to the outer surface of gold NPs, and mice immunized with Cap-AuNP showed that Cap-AuNP could efficiently activate T lymphocytes and balance the immune response of Th1 and Th2 cells (*Ding et al., 2017b*). Recently, researchers have developed a variety of gold nano-vaccines against influenza viruses that are associated with favorable application prospects (*Tao et al., 2017*; *Wang, Zhu & Wang (2017b)*). *Wang et al. (2018b)* coupled recombinant trimetric influenza A/Aichi/2/68 (H3N2) HA onto the surface of AuNPs and used flagellin as a built-in adjuvant to develop a FliC-coupled AuNP-HA nano-vaccine. These studies indicate that there is promising future for polymeric and inorganic NPs in vaccine development.

## CONCLUSIONS

With the expansion of knowledge in the fields of immunology and pathogenic biology, a new era of vaccine science has been established. Such advances have provided a basis for the development of various epitope-based vaccines that have been extensively studied due to their unique advantages, particularly the ability to overcome the safety problems associated with traditional vaccines. To overcome the obstacles associated with epitope-based vaccines (e.g., enzymatic degradation), facilitate recognition by the target cells of the immune system more efficiently, as well as maintain and enhance the efficiency and immunoreactivity of the constructed vaccines, the development of novel built-in adjuvants is a key step in the design of epitope-based vaccines. It is essential to optimize the subunit vaccine antigen design and understand the trends in adjuvant applications, including target antigen processing and its in vivo presentation. To efficiently separate the constructed multi-epitope domains, flexible linkers are often used (e.g., GPGPG and EAAK). The lack of suitable linkers in epitope-based vaccines may lead to the production of novel structural regions which may interfere with the immunogenicity of exogenous epitopes. In this review, we briefly introduced several commonly used built-in adjuvants (e.g., TLR ligands, VLPs, and several bacterial toxin proteins) with different features and several new potential co-delivery systems for epitope-based antigens. These systems are capable of forming NPs, have no immunogenicity or toxicity, and can display antigen epitopes on the surface of the particles, including MAP, SAPNs, LCP, and polymeric or inorganic nanomaterials. Additionally, carrier proteins with epitopes can serve as chaperones and stimulate the production of immune-related factors, possess CTL and Th-cell epitopes required to generate a humoral and cellular immune response, or spontaneously assemble into VLPs. The majority of built-in adjuvants (e.g., HSP70, flagellin, and Chitosan) mentioned in this review are already in clinical trials (Table 5). The carbohydrate-based conjugate vaccines were prepared using TT, DT, meningococcus B outer membrane protein complex (OMPC), and other proteins (e.g., KLH) used as the carrier molecules (*Pichichero, 2013*). Additionally, these protein carriers have achieved great success. For example, a range of glycoconjugate vaccines against infectious diseases (e.g., *Haemophilus influenzae* type B, *Neisseria meningitidis* and *pneumonia*) have been licensed for clinical use (*Astronomo & Burton, 2010*). There are some approved VLP-based vaccines currently on the market, including recombinant HBV, hepatitis E virus, and HPV vaccines. Several VLP vaccines for different diseases are

also being tested in clinical trials (*Negahdaripour et al., 2017b*). However, the clinical development of HBc-based VLPs for prophylactic vaccines is likely to experience little growth in the next few years. This is because for HBV-infected patients, the use of HBV particles as an immunogen may lead to poor responses. Thus, it is particularly important to develop alternative VLPs, such as hepadnaviral or plant virus-derived VLPs (e.g., TMV-based VLPs), some of which could be tested in future clinical trials (*Roose et al., 2013*). Although some adjuvants are currently used in vaccines licensed for human use, they are usually used as mixtures with antigens. In comparison, antigen-adjuvant fusion can significantly improve immunogenicity and display greater potential to induce an antigen-specific immune response. Based on clinical studies of conjugate vaccines, it is difficult to conclude which built-in adjuvant has a greater influence on vaccine immunogenicity. This is because, in addition to adjuvants and selected epitopes, there are other parameters that affect the immunogenicity of epitope vaccines, including conjugation chemistry, the presence of a spacer, and degree of conjugation. The development of future multivalent epitope-based vaccines is primarily dependent on future advances in research involving built-in adjuvants, which have the characteristics of being easily obtained, good biological safety, and a high efficiency for displaying epitopes. In the future, we hope to see the marketing approval of several multiepitope-adjuvant fusion vaccines, as well as increased interest in the field of built-in adjuvants.

### Funding

This work was supported by the National Key Research and Development Program (2017YFD0500902), the Key R&D Program of Gansu Province of China (Grant No. 17YF1NA070) and the National Pig Industrial System of China (CARS-36-06B). The funders had no role in study design, data collection and analysis, decision to publish, or preparation of the manuscript.

### Grant Disclosures

The following grant information was disclosed by the authors:
National Key Research and Development Program: 2017YFD0500902.
Key R&D Program of Gansu Province of China: 17YF1NA070.
National Pig Industrial System of China: CARS-36-06B.

### Competing Interests

The authors declare that they have no competing interests.

### Author Contributions

- Yao Lei conceived and designed the experiments, performed the experiments, analyzed the data, contributed reagents/materials/analysis tools, prepared figures and/or tables, authored or reviewed drafts of the paper, approved the final draft.
- Furong Zhao performed the experiments, contributed reagents/materials/analysis tools, authored or reviewed drafts of the paper.

- Junjun Shao performed the experiments, contributed reagents/materials/analysis tools.
- Yangfan Li performed the experiments.
- Shifang Li performed the experiments.
- Huiyun Chang conceived and designed the experiments, authored or reviewed drafts of the paper, approved the final draft.
- Yongguang Zhang conceived and designed the experiments, approved the final draft.

## Data Availability

This is a review article and did not generate raw data.

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
