# Peer review of "Application of built-in adjuvants for epitope-based vaccines"

_PeerJ, doi:10.7717/peerj.6185_

## Round 0.1 · original submission · Minor Revisions

Your work has been reviewed by 4 experts in the field. Even though they seems to suggest "major revision", after carefully reading their comments, I think that they can be considered mostly as "minor" and therefore I invite you to modify accordingly.

Reviewer 1 ·

Basic reporting

NO comments

Experimental design

Authors should mention the period of time selected for the references

Validity of the findings

NO comments

Additional comments

In general, the article is well written. the relevance of the references is clearly identified and the order is understandable. However, I consider that a broad section should be added and should include why most of the described adjuvants have not gone through clinical trials?, which are the biggest limitations? A deeper analysis of them with possible alternatives are necessary .

It would be interesting to include a table indicating the test phase in which each of them is located, from preclinical to clinical phases.

Reviewer 2 ·

Basic reporting

This review is broad and cross-disciplinary interest. The authors present detailed review on adjuvants and vaccine strategies.

Experimental design

Survey Methodology is consistent with a comprehensive coverage of the vaccine immunology.

Validity of the findings

Conclusion addresses gaps in knowledge and future directions.1. Please discuss protective antigen of B. anthracis which can also be used as a cytosolic delivery agent of polypeptides, nucleic acids and other noncanonical biomolecules into the cell. Please refer PMID: 28137237.
2. Line 228 “In addition, this…. No allergic reactions to TLR receptor expression on the surface of immune cells”. I don’t understand do you mean allergic reaction to immune cells?
3. Since the authors covered Tuberculosis, the immune response to antigens such as ESAT-6 and Mycobacterial lipid antigens should also be mentioned Please refer PMID: 28137237, 12882829, 17351619

4. Please look for the formatting and English mistakes. I will cite few example.
a. Line 68-70: This sentence is wrong. Thus, an…….adaptive immune response”
b. Line 89-92 “These epitopes ….their surface”. Epitopes composed of B- cell ? Please reframe sentence.
c. Line 150.. “David et al…” Reference is wrong.
d. Line 205 “chemokines(Fu et……” and Line 244 “activity (Deng…”Please check formatting.
e. Line 222 “production of INF-…..” It should be IFN
f. Font title and all Capital letters at some place.
g. Line 374 Heat Labile toxin is referred as LT. LT is often used for lethal toxin of B. anthracis. Please change the abbreviation.
h. Line 446 “The immunogenicity of adjuvanted and non-adjuvanted”. I am not sure adjuvanted is correct term here. Please restructure sentence.
i. Line 278-279 “HBcAg can……epitopes” Sentence is not clear.
j. Figure 2 legends : mediates the production of type I IFNs, such as IFN a/b/g.
k. Table 1: “Described above” Please describe properly. Also add a column for references.

5. Please include chemical structure/formula of different Pam lipopeptides in table.

6. Table 3. Please include citation of original references for MALP2(PMID:
9166424). Please check other places too.

·

Basic reporting

The article is well written and grammatically acceptable. However the authors have failed to add many recent references in the area which will be very beneficial to the readers. Further, the manuscript lacks the details about the mechanistic action of various adjuvants. The reference styling is good and the tables and figures are drawn right. However, there is no labelling and captioning of the diagrams. The authors shall also cite the sources where from the figures have been taken and wherever required, the necessary permissions must be sought.

Experimental design

The authors need to study further the left over classes of adjuvants and to add those with proper citations.

Validity of the findings

The work is relevant and meaningful addition to already existing literature.

Additional comments

The paper should be revised thoroughly revised with the above considerations in mind.

Reviewer 4 ·

Basic reporting

no comment

Experimental design

no comment

Validity of the findings

no comment

Additional comments

Lei et al., review the recent literature related to adjuvants for peptide vaccines focusing in 4 general groups: Pater Recognition Receptors, Virus Like Particles platforms, Bacterial Toxins and other organic and inorganic nanoparticles. The review is of general interest for life sciences scientist and academics (and in particular, to immunologist and vaccinologist), to introduce and broadly inform on this particular area of the adjuvants field. I have not found similar reviews at least in the past 2 years; I therefore consider this article will be a useful update in the field. The article is clearly written, figures and tables are relevant to the content of the article. Most of relevant prior literature is cited, however, I consider review will importantly benefit from including other important adjuvants that have been important to develop vaccines against several virus and bacterial induced diseases. For the section of PRRs agonists, the inclusion of adjuvants based on Outer Membrane Proteins from Gram-negative bacteria (i.e. Neisseria, Salmonella) will be of interest. For the VLP section, the inclusion of plant virus VLP platforms (i.e. Tobacco and Papaya Mosaic Virus) will enrich the review. In addition, the inclusion of outer membrane vesicles and Generalized Modules for Membrane Antigens (GMMA) will also benefit the text. Finally, I recommend the inclusion of chitosan as adjuvant in the section of organic and inorganic nanoparticles.
Please check references of Azmi F., on line 569 and 571 seems repeated.

---

## Round 0.2 · Minor Revisions

The manuscript is almost Accepted. I am returning it to you to give you an opportunity to correct the references as suggested by Reviewer 2, and to give you the opportunity to include a few sentences about other platforms as suggested by the same reviewer.

Reviewer 2 ·

Basic reporting

The article is written in clear and unambiguous, professional English used throughout. Literature is well referenced.

Experimental design

Sources are adequately cited.

Validity of the findings

Conclusion are well stated and provide future directions.

Additional comments

I will like to congratulate authors on doing commendable job in revising manuscript. I am overall satisfied with the current version.

Reviewer 4 ·

Basic reporting

no comment

Experimental design

no comment

Validity of the findings

no comment

Additional comments

Authors have properly responded to almost all my previous suggestions, the manuscript is importantly improved, however, is still missing the brief description of Papaya Mosaic Virus vaccine adjuvant platform and the description of Salmonella porins as adjuvants/immunomodulators as requested previously. (in particular the description of Salmonella porins used as adjuvant will strength the message for figure 6). In addition, the repeated references in the manuscript have not been corrected:
771 Azmi, F., Ahmad, F.A., Skwarczynski, M., and Toth, I. 2014a. Recent progress in adjuvant discovery for peptide-
772 based subunit vaccines. Hum Vaccin Immunother 10:778-796.
773 Azmi, F., Ahmad, F.A., Skwarczynski, M., and Toth, I. 2014b. Recent progress in adjuvant discovery for peptide-
774 based subunit vaccines. Hum Vaccin Immunother 10:778-796.

I consider these will help to further improve this review and extend its broad and cross-disciplinary interest.

---

## Round 0.3 · accepted · Accept

Thank you for taking into account all comments, and modified your work accordingly.

#